# CRISPR editing of *sftb-1/SF3B1* in *Caenorhabditis elegans* allows the identification of synthetic interactions with cancer-related mutations and the chemical inhibition of splicing

Xènia Serrat[1], Dmytro Kukhtar[1], Eric Cornes[2], Anna Esteve-Codina[3,4], Helena Benlloch[1], Germano Cecere[2], Julián Cerón[1]*

**1** Modeling human diseases in *C. elegans* Group, Genes, Disease and Therapy Program, Institut d'Investigació Biomèdica de Bellvitge–IDIBELL, Barcelona, Spain, **2** Mechanisms of Epigenetic Inheritance, Department of Developmental and Stem Cell Biology, Institut Pasteur, UMR3738, CNRS, Paris, France, **3** CNAG-CRG, Centre for Genomic Regulation (CRG), The Barcelona Institute of Science and Technology, Barcelona, Spain, **4** Universitat Pompeu Fabra (UPF), Barcelona, Spain

* jceron@idibell.cat

**Data Availability Statement:** The RNA sequencing data generated in this study have been deposited in

## Abstract

*SF3B1* is the most frequently mutated splicing factor in cancer. Mutations in *SF3B1* likely confer clonal advantages to cancer cells but they may also confer vulnerabilities that can be therapeutically targeted. *SF3B1* cancer mutations can be maintained in homozygosis in *C. elegans*, allowing synthetic lethal screens with a homogeneous population of animals. These mutations cause alternative splicing (AS) defects in *C. elegans*, as it occurs in *SF3B1*-mutated human cells. In a screen, we identified RNAi of U2 snRNP components that cause synthetic lethality with *sftb-1/SF3B1* mutations. We also detected synthetic interactions between *sftb-1* mutants and cancer-related mutations in *uaf-2/U2AF1* or *rsp-4/SRSF2*, demonstrating that this model can identify interactions between mutations that are mutually exclusive in human tumors. Finally, we have edited an SFTB-1 domain to sensitize *C. elegans* to the splicing modulators pladienolide B and herboxidiene. Thus, we have established a multicellular model for *SF3B1* mutations amenable for high-throughput genetic and chemical screens.

## Author summary

*SF3B1* encodes a splicing factor frequently mutated in hematological malignancies, and with less frequency in other solid tumors. Each *SF3B1* mutation is predominant in a cancer type, indicating cell type-specific pathogenic mechanisms. We have reproduced three cancer-related *SF3B1* missense mutations in the *C. elegans* ortholog *sftb-1* and they do not cause any overt phenotypes, but few alterations in splicing. However, the combination of these mutations has additive effects, hampering SFTB-1 functions and allowing the study of synthetic interactions that may result in a defective spliceosome structure. As a result of

the Gene Expression Omnibus repository and are available under the accession number GEO: GSE129642. All other relevant data are within the manuscript and its Supporting Information files.

**Funding:** This work has been supported by a grant from the Instituto de Salud Carlos III (ISCIII), http://www.isciii.es/, to JC (PI15-00895), co-funded by FEDER funds/European Regional Development Fund (ERDF) — a way to build Europe, https://ec.europa.eu/regional_policy/en/funding/erdf. AEC is funded by ISCIII, http://www.isciii.es/, of the MINECO (reference PT17/0009/0019) and co-financed by FEDER, https://ec.europa.eu/regional_policy/en/funding/erdf/. EC is supported by a Pasteur-Roux postdoctoral fellowship program, https://research.pasteur.fr/en/call/call-for-application-pasteur-roux-and-pasteurcantarini-postdoctoral-fellowships-2019-spring-session/. XS has an FPU PhD fellowship from MINECO, http://www.educacionyfp.gob.es/servicios-al-ciudadano-mecd/becas-ayudas.html. DK has an FI AGAUR fellowship from Generalitat de Catalunya, http://agaur.gencat.cat/en/inici/. The funders had no role in study design, data collection and analysis, decision to publish, or preparation of the manuscript.

**Competing interests:** The authors have declared that no competing interests exist.

an RNAi screen, we identified other proteins of the U2 snRNP complex as targets for synthetic lethality. The *sftb-1* mutant strains also allow the identification of mutations in other splicing factors that are mutually exclusive with *SF3B1* mutations in tumors. *SF3B1* cancer mutations are heterozygous, and interestingly, the most promising therapeutic venue is based on pladienolide B (PB) derivatives that hit the wild-type and the mutated protein. We have edited SFTB-1 to sensitize it to pladienolide B and herboxidiene (HB), therefore making *C. elegans* suitable for assaying the toxicity and efficacy of their chemical derivatives. Thus, our model represents a pre-clinical system for investigating both synthetic lethality and chemotherapies. Besides that, we have generated the first *C. elegans* strain in which splicing can be chemically modulated by PB and HB.

## Introduction

Deregulated RNA splicing is emerging as a new hallmark of cancer following the discovery of several splicing factor genes harboring somatic mutations in different tumor types [1]. *SF3B1*, a core component of the U2 snRNP, is the most frequently mutated splicing factor in human cancers. Somatic heterozygous mutations in *SF3B1* are particularly prevalent in myelodysplastic syndromes (MDS)—up to 80% in refractory anemia with ring sideroblasts [2–4]; and in 15% of chronic lymphocytic leukemia (CLL) [5–7]. *SF3B1* mutations have also been reported in solid tumors, including 20% of uveal melanomas (UM) [8–10], 3% of pancreatic ductal adenocarcinomas [11], and 1.8% of breast cancers [12]. RNA-sequencing (RNA-seq) analyses in different tumor types and cell lines with *SF3B1* missense mutations, including the most prevalent substitution K700E, identified distinct alternative splicing (AS) defects [13–15].

Recently, CRISPR/Cas9 genome editing has allowed researchers to faithfully reproduce human pathological mutations in animal models. Human *SF3B1* mutations have been introduced in cell lines [15–18], but mimicking these mutations in multicellular organisms has only been achieved by conditional alleles in murine models [19]. Taking advantage of the extraordinary conservation of splicing factors across evolution and the ease of genetic manipulation of *C. elegans*, we established a multicellular model to study *SF3B1* cancer-related mutations. Spliceosome components, and particularly *SF3B1*, are being intensively studied as targets of antitumor drugs [20–22]. Some natural compounds modulate splicing by targeting SF3B1. Among them, pladienolide B is particularly relevant because H3B-8800, a PB derivative, has antitumoral effects and is in clinical trials [23]. The modulatory activity of PB is highly dependent on SF3B1 structure, and single amino acid substitutions cause resistance to PB [24,25]. Hence, particular residues in the drug binding site that are not conserved in SFTB-1 may confer resistance to PB in worms, as it has been recently observed in yeast [26]. We edited four amino acids to humanize the HEAT repeat 15 in SFTB-1, making *C. elegans* sensitive to splicing modulators. Thus, we have created the first *C. elegans* strain sensitive to PB and HB, which would facilitate the chemical modulation of splicing in this model system.

Altogether, the multicellular model of *SF3B1* mutations described in this study provides a new pre-clinical platform for identifying new targets and small molecules for use in cancer therapies.

## Results

### *sftb-1*, the *C. elegans* ortholog of *SF3B1*, is a ubiquitously expressed gene essential for development

The *C. elegans* protein SFTB-1 is 66% identical to human SF3B1 in terms of amino acid composition. Homology is particularly high at the HEAT domain, reaching 89% identity, and the

most frequently mutated amino acids in cancer are conserved. The SFTB-1 sequence also conserves some of the U2AF ligand motifs (ULMs) that bind U2AF homology motifs (UHMs) present in other splicing factors (Figs 1A and S1) [27].

As expected for a core splicing factor, the CRISPR-engineered endogenous fluorescent reporter mCherry::SFTB-1 showed ubiquitous expression in somatic and germ cells, being absent in mature sperm only (Fig 1B and 1C). We also used CRISPR to generate the *sftb-1 (cer6)* mutation, a deletion allele that produces a premature stop codon (S2 Fig) and causes an arrest at early larval stages, confirming that *sftb-1* is essential for development (Fig 1D and 1E). *sftb-1(cer6)* homozygous animals completed embryonic development but arrested as larvae that could undergo few germ cell divisions (S2 Fig). Since embryos depleted of *sftb-1* transcripts are not viable [28], our observation indicates that maternal wild-type (WT) product can still be present in the early larva. We formally tested this hypothesis by combining the *sftb-1* deletion allele and the *sftb-1* endogenous mCherry reporter in the same strain (S2 Fig). The compound heterozygotes produced a normal brood size and segregated 19.39% non-red *cer6* homozygotes (S2 Fig). By monitoring mCherry::SFTB-1 expression in the progeny of heterozygous mothers, we observed a small percentage of early L1 larvae with dim red fluorescent expression that later became non-red and developmentally arrested larvae (S2 Fig), indicating that there is some SFTB-1 maternal product reaching the early L1 stage.

## Different *sftb-1* missense mutations display additive effects when combined

We edited the *C. elegans* genome to mimic the K700E mutation, the most frequent *SF3B1* substitution in tumors. *sftb-1*[K718E] homozygous animals did not display any obvious phenotypes. Thus, we decided to reproduce two other *SF3B1* cancer-related mutations. R625C is a particularly prevalent mutation in uveal melanoma, and Q534P is less common but is predicted to have a strong impact on SF3B1 structure due to its location in an α-helix [29]. Similar to *sftb-1*[K718E], the two additional missense mutations, *sftb-1*[R643C] and *sftb-1*[Q552P] (S3 Fig), did not cause any evident phenotypes in *C. elegans*.

We were interested in producing animals with defective *sftb-1* function resulting in a viable and traceable phenotype for investigating modifiers of altered SFTB-1 activity. Since the effects of MDS-related missense mutations were reported to be additive in the yeast *SF3B1* ortholog Hsh155 [30], we generated three strains with each of the possible pairs of mutations: *sftb-1* [R643C, K718E], *sftb-1*[Q552P, K718E], and *sftb-1*[Q552P, R643C]. None of the three double mutants showed an obvious phenotype. However, animals simultaneously carrying the three missense mutations *sftb-1*[Q552P, R643C, K718E] displayed several temperature-sensitive developmental defects including partial sterility, protruding vulva, and developmental delay (Figs 1F–1H and S3). These results suggest that each of the three missense mutations provokes a functional impact that becomes unmasked only when the three mutations are combined.

## Distinct *sftb-1* alleles produce alterations in alternative splicing and gene expression

The most commonly described AS defect caused by cancer-associated *SF3B1* mutations is the incorrect recognition of 3' splice sites, producing aberrant transcripts [13,14,18,31,32]. We investigated in *C. elegans* the transcriptional consequences of the K718E mutation, an equivalent of the most frequent *SF3B1* mutation K700E, by RNA-seq of two different strains with two independent K718E alleles (*cer3* and *cer7*) and their WT siblings at the L4 stage. Then, we used rMATS for robust detection of differential AS events between mutant and WT worms [33]. Using both splice junction and exon body reads, our analysis detected 21,985 AS events, of which 134 were significantly altered (FDR < 0.05) (S1 Table). A more restricted cut-off

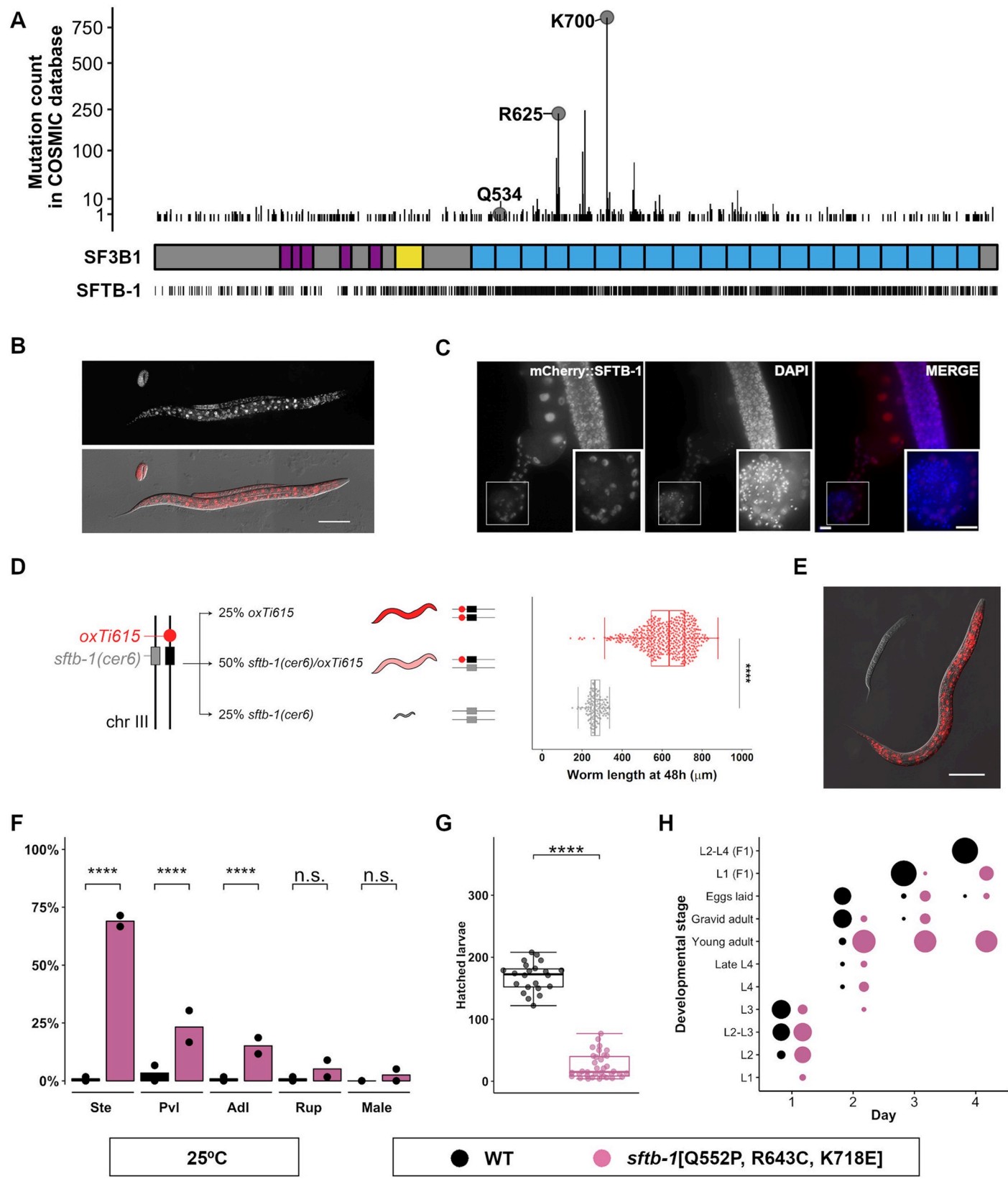

**Fig 1. Characterization of *sftb-1*, the *C. elegans* ortholog of *SF3B1*, and different *sftb-1* mutations.** (A) Number of *SF3B1* missense mutations annotated in COSMIC database [75] at each amino acid position, with labels placed on the relevant residues for this study. A schematic structure of the SF3B1 protein is shown below, where purple, yellow, and blue boxes represent the U2AF ligand motifs (ULMs), the p14-interacting region, and the 20 HEAT repeats, respectively. The bars at the bottom line indicate residues that are conserved in the *C. elegans* protein. (B) Representative image of an embryo, L2, and L4 larvae with mCherry::SFTB-1 endogenous expression. The fluorescence image (top panel) is merged with the corresponding differential interference contrast (DIC) image (bottom panel). (C) Fluorescence images of dissected gonads from adult worms expressing mCherry::SFTB-1. Gonads were stained with DAPI to visualize nuclei. Insets show differentiated sperm nuclei, where mCherry::SFTB-1 is not expressed. (D) Left: diagram depicting how the *cer6* deletion allele is maintained in heterozygosis. The WT *sftb-1* copy is genetically linked to a red fluorescent marker (*oxTi615*). Right: worm body length of WT and heterozygotes (red, n = 475), and *cer6* homozygotes (gray, n = 134) grown for 48 h at 20˚C (N = 3). (E) Representative image of synchronized red and non-red siblings after growing for 48 h at 20˚C. (F) Incidence of different *sftb-1*[Q552P, R643C, K718E] phenotypes at 25˚C. Dots represent percentages observed in each replicate (n = 119, 116; N = 2). (G) Brood size of WT or *sftb-1* [Q552P, R643C, K718E] fertile worms at 25˚C (n = 22, 35; N = 2). (H) *sftb-1*[Q552P, R643C, K718E] animals display developmental delay at 25˚C. The proportion of the population at each stage is represented by the size of the dots (n = 119, 116; N = 2). In (D) and (G), dots represent measures in individual worms, and overlaid Tukey-style boxplots represent the median and interquartile range (IQR). Whiskers extend to the minimum and maximum values or at a maximum distance of 1.5xIQR from the box limits. Statistics: (D) and (G), Mann-Whitney's test; (F), Fisher's exact test. n.s., no significant difference, **** p<0.0001. Scale bars: 100 μm (B and E), 10 μm (C).

(inclusion level difference > 0.1), similar to that used in a previous rMATS analysis of *SF3B1*-mutated samples [32], reduced the list of significant AS events to 78, with skipped exon (SE) being the most frequent event (Fig 2A and 2B).

Then, we investigated whether the two *sftb-1* alleles associated with a visible phenotype (the triple mutation allele *cer39*, and the deletion allele *cer6*) primarily induced cryptic 3' splice site selection. It is worth noting that, in this experiment, larvae were harvested and processed for RNA isolation at the L2 stage to avoid possible indirect effects due to developmental defects observed after this stage. Both alleles were associated with a high number of deregulated AS events as determined by rMATS, and the predominant alteration was also exon skipping (Fig 2B).

We then explored if there were differentially expressed protein-coding (Fig 2C) and non-coding (Fig 2D) transcripts in the three RNA-seq datasets (S2 Table). As expected, the highest number of significantly altered transcripts was observed in the *cer6* allele.

A functional enrichment analysis uncovered biological pathways that were significantly affected by the distinct *sftb-1* alleles (S4 Fig). The K718E mutation caused the downregulation of transcripts related to mitochondrial functions, which were also downregulated by the deletion allele *cer6*. Both *cer39* and *cer6* alleles induced the upregulation of transcripts involved in mRNA processing, and the downregulation of different metabolic pathways. Thus, it seems that the common reaction to an impaired splicing is to increase the splicing components and reduce the metabolism.

Interestingly, there is a partial overlap between the transcripts deregulated by the triple mutation allele *cer39* and the deletion allele *cer6* (Fig 2E). We also found few AS events that are similarly altered in distinct *sftb-1* mutants. The largest overlap was again observed between the *cer39* and the *cer6* alleles, which shared 80 deregulated events (S5 Fig). Thus, the additive effect of three missense mutations produces a similar impact on gene expression as the deletion allele but with a considerable number of unique AS changes.

In summary, we identified gene expression and AS alterations that are more frequent as *sftb-1* function becomes progressively compromised, with SE being the predominant AS defect.

## A subset of U2 snRNP and U2-associated components synthetically interacts with *sftb-1* mutants

In the search for selective vulnerabilities in *sftb-1* defective backgrounds, we hypothesized that *sftb-1* mutant animals would be more sensitive to knockdown of other splicing factors than their WT counterparts. Based on a previous *C. elegans* RNA interference (RNAi) collection of

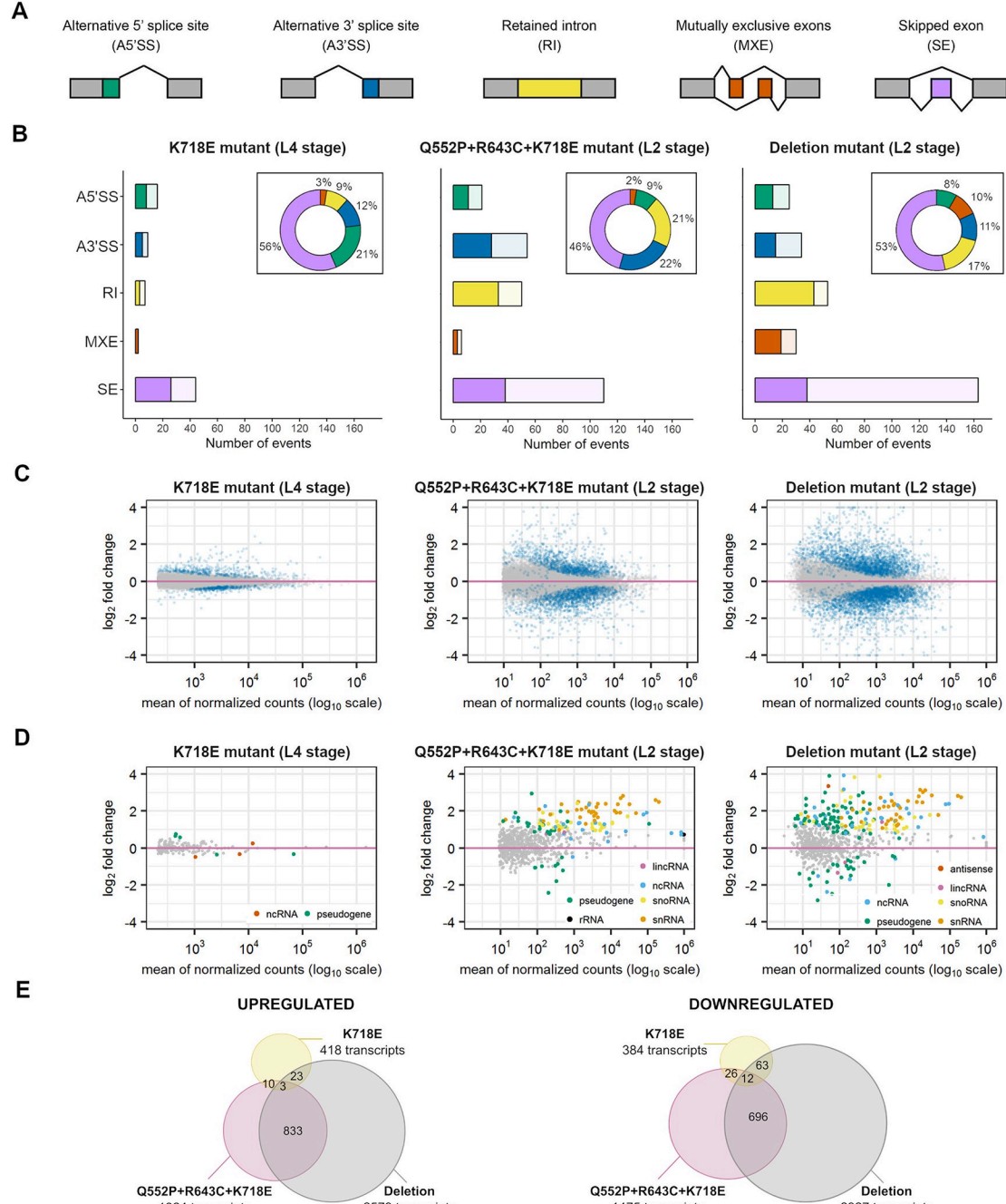

**Fig 2. Different *sftb-1* mutations induce alternative splicing and gene expression alterations.** (A) Types of AS events. Gray boxes represent constitutive exons, while colored boxes represent alternatively spliced exons. (B) Total number of significant (FDR<0.05) AS alterations detected by rMATS with an inclusion level difference > 0.1. Event types are color coded as in panel A. Light colors represent AS events that were significantly more included in WT samples, while dark colors symbolize splicing alterations associated to mutant samples. Inset doughnut charts show the distribution of significant events by event type. In the three mutants, skipped exon (SE) was the most predominant alteration. (C) MA-plots displaying changes in protein-coding transcript expression levels (corrected log$_2$ fold change) over the mean of normalized counts, analyzed by DESeq2. Significant transcripts (padj<0.05) are colored in blue. (D) MA-plots of expression changes in non-coding transcripts, represented as in panel C. Transcripts that are significantly deregulated are colored by transcript type. (E) Venn diagrams showing the overlap between transcripts that are upregulated (left) or downregulated (right) in the three datasets. The number of transcripts at the intersections is shown, as well as the total number of significantly up- or downregulated transcripts in each mutant strain (padj<0.05).

splicing factors [34], we compiled and assayed a total of 104 RNAi clones targeting proteins that act in different steps of the splicing reaction. This led us to identify 27 splicing factors whose knockdown phenotype was enhanced in at least one of the three *sftb-1* single mutants, suggesting possible synthetic interactions. Among the potential candidates, we identified proteins present in pre-catalytic and catalytic spliceosomes, as well as few auxiliary factors, indicating that *sftb-1* missense mutations might impact other steps of the splicing reaction besides early spliceosome assembly (S3 Table).

We selected three U2 snRNP components, including *sftb-1*, and two U2-associated proteins for further validation based on their presence in the U2 snRNP and the phenotypes observed (Fig 3A). The identification of *sftb-1* as a candidate certified our approach since it demonstrated that any RNAi affecting *sftb-1*-related functions would have a stronger effect on *sftb-1* mutants. Since a partial loss-of-function mutant for the candidate *teg-4* was available, we proceeded to validate the interaction between *sftb-1* mutations and *teg-4*, the *C. elegans* ortholog of *SF3B3*, which physically interacts with SF3B1 HEAT repeats 3, 5, and 6 [29]. We found that the K718E mutation exacerbated the phenotype of *teg-4(oz210)* at 20°C [35], whereas the Q552P mutation did so to a lesser extent (Fig 3B).

Then, we validated the rest of the candidates using RNAi. As we had observed that *sftb-1* mutations had an additive effect (Fig 1F–1H), we reasoned that *sftb-1* double mutants might be more sensitive to splicing perturbations than single mutants. Consistently, *sftb-1(RNAi)* led to an additive loss of function in double mutants, which arrested at earlier stages compared to WT worms (Fig 3C). A similar result was obtained with knockdown of *uaf-2*/U2AF1 (Fig 3D), another candidate from the RNAi screen.

Among the remaining U2 and U2-associated candidates, we selected *mog-2*/SNRPA1 and *smr-1*/SMNDC1 for further validation in the double-mutation backgrounds. Strikingly, whereas RNAi of *mog-2* resulted in very low penetrant sterility in WT worms, the incidence of the phenotype incremented to over 70% in the three *sftb-1* double-mutation strains (Fig 3E). A similar synthetic phenotype was observed upon knockdown of *smr-1* (Fig 3F). Finally, *sftb-1* double mutants were more sensitive to *teg-4(RNAi)* (Fig 3G). This former synthetic interaction was not as strong as the one observed with *mog-2(RNAi)* and *smr-1(RNAi)* but was supported by the interaction between genetic alleles (Fig 3B).

We also found that heterozygous animals carrying the deletion allele *sftb-1(cer6)* were sensitive to RNAi of the five interactors (S6 Fig). Still, double mutant strains are better than *cer6* heterozygous animals for identifying genetic interactions (Figs 3 and S6).

Together, our results suggest that targeting other splicing factors such as *SF3B3* to compromise their function can be an opportunity to selectively impair cancer cells harboring *SF3B1* mutations.

## The *C. elegans* mutations equivalent to human *U2AF1*[S34P] and *SRSF2*[P95H] genetically interact with *sftb-1*[Q552P, R643C, K718E] mutants

Along with *SF3B1* mutations, recurrent missense mutations in U2 small nuclear RNA auxiliary factor 1 (*U2AF1*) and serine/arginine-rich splicing factor 2 (*SRSF2*) were discovered in hematological malignancies [2], and later reported to be present in some solid tumors [36,37]. Somatic hotspot mutations in these three splicing factors are only found in heterozygosis and are mutually exclusive, suggesting redundant functional impact or limited tolerance to splicing perturbations.

Missense mutations in *U2AF1* are commonly found at position S34, which corresponds to a residue located within the two zinc finger domains through which U2AF1 interacts with the pre-mRNA [38]. Similarly, *SRSF2* is typically mutated at position P95, mainly bearing missense mutations but also in-frame insertions or deletions [2].

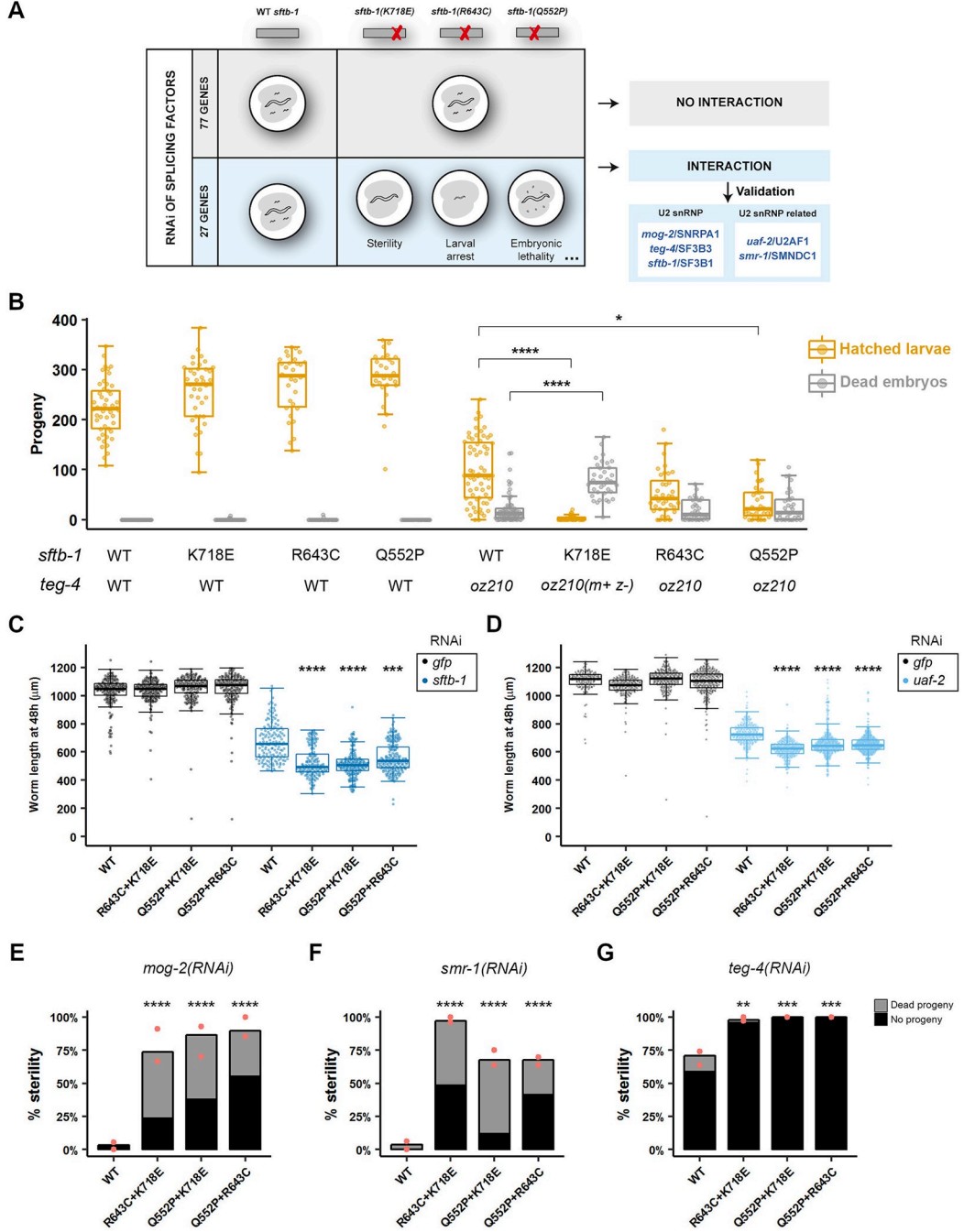

**Fig 3. Synthetic interaction between *sftb-1* mutant strains and knockdown of U2 snRNP and U2 snRNP-related components.** (A) Schematics of the RNAi screen. In brief, WT or mutant worms bearing any of the three *sftb-1* single mutations were fed with RNAi clones against a total of 104 splicing factors. 77 clones had the same effect in all genetic backgrounds. In contrast, we identified 27 clones that caused a stronger phenotype in at least one of the three single mutant strains compared to WT, which was indicative of a synthetic interaction. Listed are the 3 U2 snRNP and 2 U2 snRNP-related candidates from the screen that were selected for further validation. (B) *sftb-1*[K718E] and *sftb-1*[Q552P] synthetically interact with *teg-4(oz210)*. The number of hatched larvae (orange) and dead embryos (gray) laid by worms of the indicated genotype at 20˚C is represented (n≥28; N = 3). *m+ z-*, homozygous mutant progeny of heterozygous mothers. No significant differences were observed in the number of hatched larvae laid by *sftb-1*[K718E], *sftb-1*[R643C], and *sftb-1* [Q552P] compared to WT. (C) Worm length after 48 h of RNAi treatment against *gfp* (black) or *sftb-1* (blue) at 25˚C (n≥150; N = 2). (D) Worm length after 48 h of RNAi treatment against *gfp* (black) or *uaf-2* (blue) at 25˚C (n≥150; N = 2). In both (C) and (D), worm length was not significantly different between *sftb-1* mutants and WT upon *gfp(RNAi)*. (E-G) Mean percentage of sterile worms observed upon RNAi of *mog-2* (E; n≥30), *smr-1* (F; n≥27) and *teg-4* (G; n≥34) at 25˚C.

Red dots indicate percent sterility observed in each replicate (N = 2). Gray bars, $P_0$-treated worms giving rise to <5 $F_1$ larvae and some dead embryos ('dead progeny' category); black bars, $P_0$-treated worms that laid neither larvae nor dead embryos ('no progeny' category). In (B), (C), and (D), data are shown as Tukey-style boxplots and overlaid dots representing measures in individual animals. Statistics: (B), (C), and (D), Kruskal-Wallis test with Dunn's multiple comparison test. Data were compared to *teg-4(oz210)* (B), or to WT animals fed with the same RNAi clone (C) and (D). (E-G), Fisher's exact test with Bonferroni correction for multiple comparison. * $p<0.05$, ** $p<0.01$, *** $p<0.001$, **** $p<0.0001$.

Based on a recently reported synthetic interaction between $Srsf2^{P95H/+}$ and $Sf3b1^{K700E/+}$ in mice [39], and on the fact that *sftb-1* mutants were more sensitive to inactivation of some splicing factors (**Fig 3**), we wondered if the combination of *sftb-1*[K718E] with mutations equivalent to $U2AF1^{S34F}$ or $SRSF2^{P95H}$ would be tolerated in *C. elegans*.

Thus, we edited the worm orthologs of these genes to generate the *uaf-2*[S42F] and *rsp-4*[P100H] mutant strains. Neither of these two mutations caused any obvious phenotypes nor dramatic changes in some AS events analyzed by RT-PCR (**S7 Fig**). In contrast to its murine counterpart $Srsf2^{P95}$, *rsp-4[P100H]* is tolerated in homozygosis [39].

We did not detect any interaction when combining each of these mutants with *sftb-1*[K718E] or the three *sftb-1* double-mutation strains. In contrast, *rsp-4*[P100H] and *uaf-2*[S42F] significantly increased the low percentage of sterile worms observed in the *sftb-1*[Q552P, R643C, K718E] background at 20˚C (**Fig 4A**). The most dramatic effect was observed by the addition of the *uaf-2*[S42F] mutation to the *sftb-1* triple-mutation strain.

As a result of these synthetic interactions, we observed masculinization of the germline (Mog) and tumorous germline (Tum) phenotypes, which are hallmarks of germlines with altered splicing activity [34]. Thus, these synthetic interactions may cause a synergistic disruption of general splicing functions, perhaps through a flaw in spliceosomal architecture, although detailed information on the molecular basis of this interaction should be obtained by deep sequencing of mutant transcriptomes.

Moreover, this combination of *uaf-2* and *sftb-1* mutations induced developmental delay at 20˚C (**Fig 4B**). These results show for the first time in an animal model a synthetic interaction between *sftb-1* and *uaf-2* missense mutations.

## A humanized HEAT repeat 15 in SFTB-1 sensitizes *C. elegans* to pladienolide B and herboxidiene

The SF3B1 C-terminal region contains 20 structural motifs known as HEAT repeats, each one formed by two alpha helices joined by a short loop. These motifs display some structural plasticity and are involved in interactions with the pre-mRNA and other proteins [29]. Specific residues within SF3B1 HEAT repeats 15, 16, and 17 (H15–H17) interact with PB [40], and some amino acids in this region are not conserved in *C. elegans* (**Fig 5A**). In order to test if these sequence differences would affect the worms sensitivity to splicing modulators, we exposed WT animals and *sftb-1* cancer-related mutants to PB and to the synthetic molecule sudemycin D6 [41], which also targets the SF3B complex. Both drugs had no visible effects when delivered in liquid medium or via microinjection (**S4 Table**). To reproduce the PB binding site in *sftb-1*, we used CRISPR to introduce four missense mutations (*cer144* allele) that mimic the human HEAT repeat 15 (**Fig 5B**). The resulting strain was apparently WT but became sensitive to PB, although the substitutions were not enough to overcome the resistance to sudemycin D6. *sftb-1(cer144)* mutant larvae exposed to 10, 50, and 100 μM PB displayed a dose-dependent growth defect (**Fig 5C**). In addition, humanization of the HEAT repeat 15 also conferred sensitivity to herboxidiene (HB) (also known as GEX1A) (**Fig 5D**), which is predicted to be accommodated in the H15-H17 pocket similarly to PB [40,42].

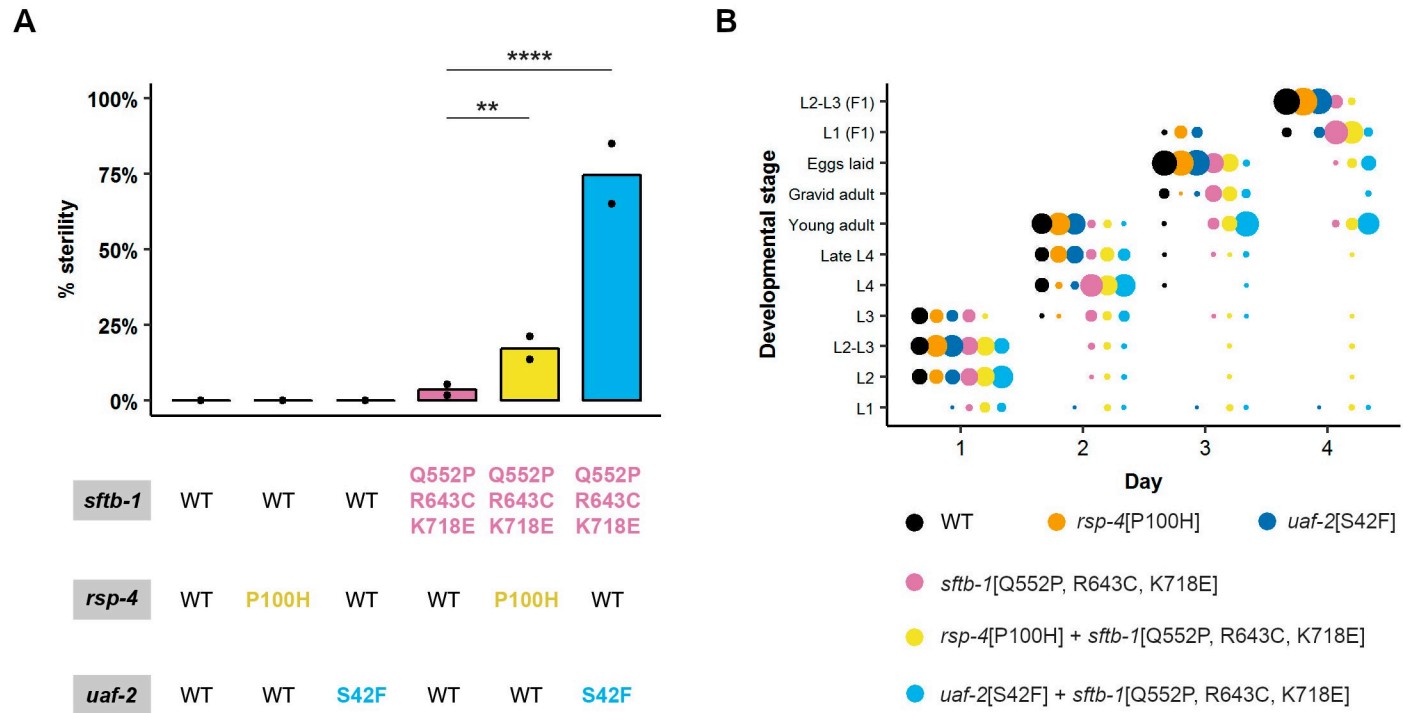

**Fig 4. Genetic interaction between *rsp-4/SRSF2*, *uaf-2/U2AF1*, and *sftb-1* mutant strains.** (A) Mean percentage of sterile worms of the indicated genotype at 20˚C (N = 2). Black dots denote observed values in each replicate. Statistical significance was determined with the Fisher's exact test with Bonferroni correction for multiple comparison. Lines indicate compared groups. ** p<0.01, **** p<0.0001. (B) *rsp-4*[P100H] and *uaf-2*[S42F] mutations gradually exacerbate the mild developmental delay displayed by the *sftb-1*[Q552P, R643C, K718E] mutant at 20˚C. Dot sizes symbolize the percentage of worms at each developmental stage on any given day. In both panels, n≥110, N = 2.

## Discussion

*SF3B1* cancer-related mutations have been intensively studied in the past few years. They are particularly recurrent in MDS or UM, which commonly lack effective therapeutic approaches. In this study, we establish the first multicellular model for *SF3B1* mutations expressed non-conditionally from the endogenous locus. The rapid *C. elegans* life cycle (three days at 25˚C) and the around 300 self-progeny obtained from a single hermaphrodite, among other features, make this model organism ideal for large-scale genetic and drug screens. The task of modeling human pathological mutations has been facilitated by the advent of the CRISPR/Cas9 genome editing technique, which has been efficiently implemented in this nematode [43,44].

As expected for a core splicing factor, and as previously reported in zebrafish [45] and mice [46], the *C. elegans SF3B1* ortholog is essential for development. Cancer cells bear *SF3B1* hotspot mutations in heterozygosis and depend on *SF3B1* WT function for viability [47]. However, homozygous *sftb-1* missense mutations are viable in *C. elegans*, although they generate defects that are unmasked when additional alterations in the spliceosome are introduced. Such a feature is advantageous for investigating vulnerabilities of *SF3B1*-mutated tumors, which are not only conferred by missense mutations, but also by the loss of one *SF3B1* WT copy [16,48].

Differently from MDS with *SF3B1* mutations, where aberrant transcripts have been directly related to phenotypic traits [32,49], transcripts affected by *sftb-1*[K718E] do not produce any obvious phenotypes. Still, $Sf3b1^{K700E/+}$ murine models only partially recapitulate the hematopoietic phenotype characteristic of MDS [50,51] and CLL [52], although a modest overlap in

**A**

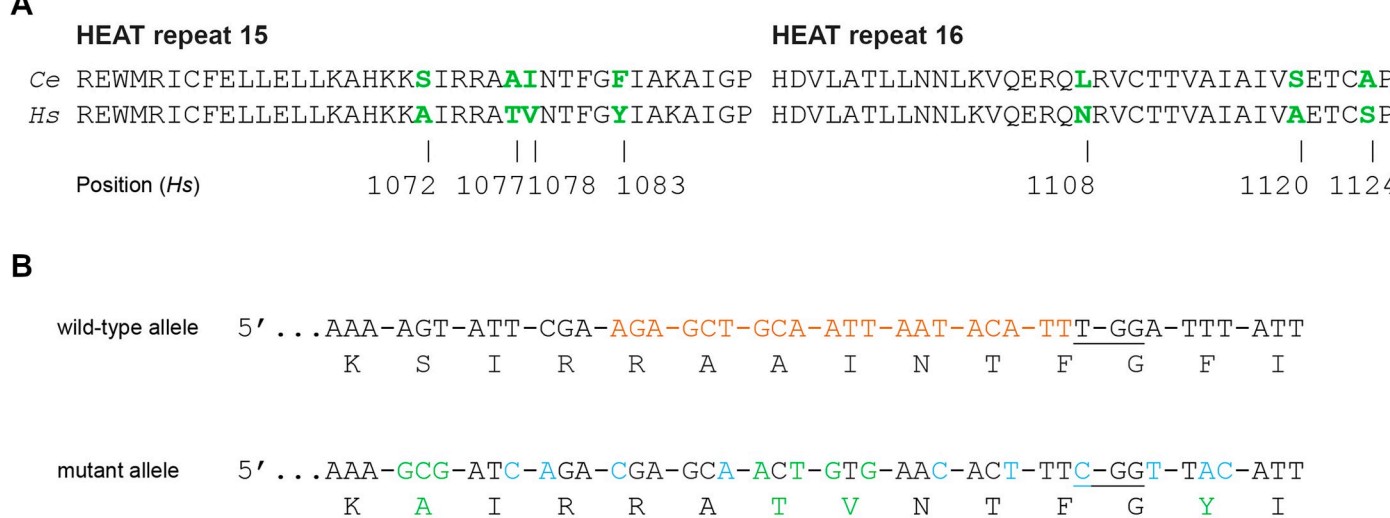

**HEAT repeat 15**

*Ce* REWMRICFELLELLKAHKK**S**IRRA**AI**NTFG**F**IAKAIGP
*Hs* REWMRICFELLELLKAHKK**A**IRRA**TV**NTFG**Y**IAKAIGP

Position (*Hs*)  1072 10771078 1083

**HEAT repeat 16**

HDVLATLLNNLKVQERQ**L**RVCTTVAIAIV**S**ETC**A**P
HDVLATLLNNLKVQERQ**N**RVCTTVAIAIV**A**ETC**S**P

1108        1120 1124

**B**

wild-type allele  5'...AAA-AGT-ATT-CGA-AGA-GCT-GCA-ATT-AAT-ACA-TT̲T-GGA̲-TTT-ATT
                      K   S   I   R   R   A   A   I   N   T   F   G   F   I

mutant allele     5'...AAA-GCG-ATC-AGA-CGA-GCA-ACT-GTG-AAC-ACT-TT̲C-GGT̲-TAC-ATT
                      K   A   I   R   R   A   T   V   N   T   F   G   Y   I

**C**

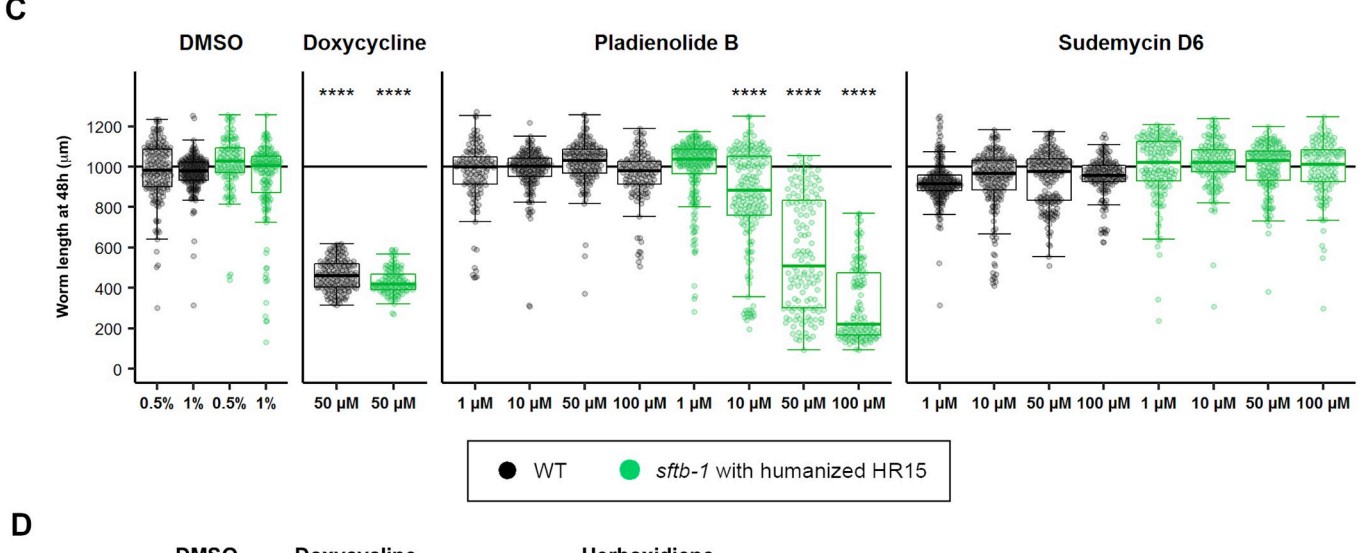

**D**

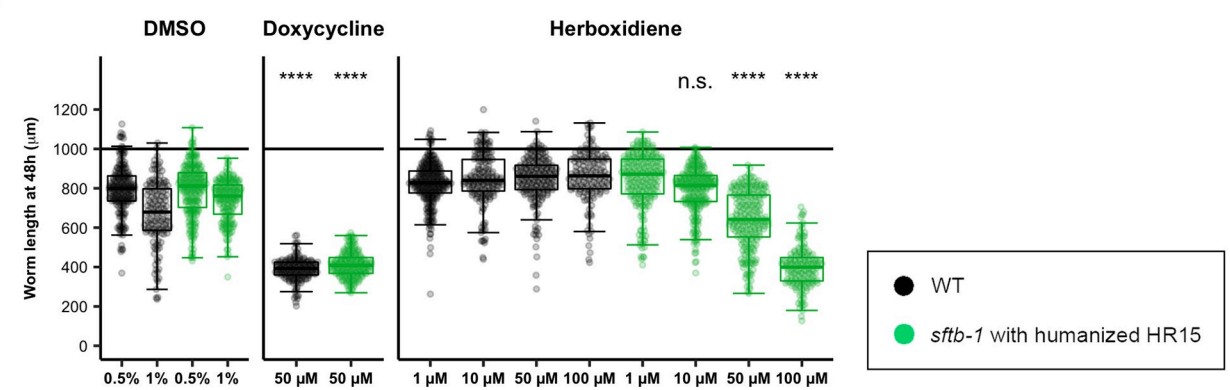

**Fig 5. Humanization of the SFTB-1 HEAT repeat 15 confers sensitivity to pladienolide B and herboxidiene.** (A) Protein sequence alignment of residues in HEAT repeats 15 and 16. Amino acids that differ between worm SFTB-1 (*Ce*) and human SF3B1 (*Hs*) are colored in green. Positions of these residues in the human sequence are indicated. (B) CRISPR/Cas9 design to "humanize" the SFTB-1 HR15. In the wild-type allele, the crRNA sequence used is indicated by orange nucleotides, and the protospacer-adjacent motif (PAM) is underlined. In the mutant ("humanized") allele, synonymous mutations introduced to prevent partial recombination events and to improve primer specificity for mutant allele detection by PCR are indicated in blue. Missense mutations and the corresponding amino acids are colored in green. (C) Worm length after 48 h of drug treatment at 25˚C (n≥120; N = 3). WT animals are represented as black dots, and *sftb-1(cer144)*

mutants as green dots. Doxycycline treatment induced larval arrest in both strains and was used as positive control. Data are shown as Tukey-style boxplots and overlaid dots represent measures in individual animals. The horizontal line at 1000 μm indicates the expected body length of an adult worm. Body length of mutant worms treated with 50 μM doxycycline and 10 μM and 50 μM PB differed statistically from that of mutant worms treated with 0.5% DMSO (p<0.0001). The same applied when comparing mutant worms treated with 100 μM PB and 1% DMSO. Significance was determined with the Kruskal-Wallis test with Dunn's multiple comparison test. (D) Worm length after 48 h of herboxidiene or doxycycline (control) treatment at 25˚C (n≥139; N = 3). Data are represented and analyzed as in panel C (n.s., no significant difference).

aberrant splicing events between $Sf3b1^{K700E/+}$ mice and MDS patients with *SF3B1* mutations was detected [39].

RNA-seq of *sftb-1*[K718E] animals did not identify the aberrant use of alternative 3' splice sites (A3'SS) as the predominant splicing defect caused by this mutation, as has been observed in human cells [13,14,16,18,31,32,52,53]. Indeed, in three distinct *sftb-1* mutants, we found a mild effect on AS, with SE being the most abundant splicing defect. Such disparity in the consequences of the K700E mutation between humans and nematodes may be explained by different factors. First, diverse computational pipelines have been used to assess differences in alternative splicing between WT and *SF3B1*-mutated samples, resulting in distinct conclusions. For instance, *Liberante* and coworkers reanalyzed previously published data and found that *SF3B1* mutation was mostly associated with differential skipped exon events, rather than A3'SS selection [15]. Second, intronic elements directing splicing have evolved differently in *C. elegans* compared to other organisms. *C. elegans* introns are typically shorter, do not have a consensus branch site (BS) sequence, and the vertebrate polypyrimidine (Py) tract is replaced by the consensus UUUUCAG/R sequence [54]. Third, only ~35% of genes are alternatively spliced in *C. elegans* [55] compared to 95% in humans [56]. Fourth, cell type-specific alternative isoforms generated by *sftb-1* mutations could have been underrepresented in our analyses as we sequenced the transcriptome of whole animals, and A3'SS selection can be regulated in a tissue-specific manner in *C. elegans* [57]. And fifth, the nonsense-mediated decay (NMD) system could efficiently eliminate transcripts with A3'SS resulting from *sftb-1* mutations, as occurs in human cells [13]. However, we did not detect predominance of downregulation among the differentially expressed transcripts (Fig 2C and 2D).

Beyond its action on A3'SS usage, *SF3B1* missense mutations could have an impact on the structure, the interaction with other proteins in the spliceosome, or the robustness of the protein. In fact, the PB derivative H3B-8800, which induces lethality in spliceosome-mutant cancers by modulating the SF3B complex, affects AS activity similarly in WT and in $SF3B1^{K700E}$ human cells [23]. Therefore, *SF3B1*-mutated cells present vulnerabilities that can be exploited in synthetic lethal screens. We found that *sftb-1* mutants are sensitive to RNAi of different splicing factors, including itself, and we further validated these synthetic interactions with the inactivation of other components of the U2 snRNP. These results point to other components of the spliceosome as targets to preferentially kill cells with deficiencies in SF3B1 activity. Knowing the suitability of *sftb-1* mutants for RNAi screens, the collection of RNAi clones to be tested should be expanded, with particular interest placed on deubiquitinases and chromatin factors that have been related to SF3B1 functions [48,58].

Pladienolide and related compounds have been proven to be effective in selectively killing cells with *SF3B1* mutations [23,50,59]. We tested pladienolide B and sudemycin D6 in WT animals and *sftb-1* cancer-related mutants but did not observe any effects (S4 Table). The modulatory activity of these drugs is highly dependent on the protein structure, as evidenced by the fact that missense mutations in *SF3B1* confer resistance to the drugs [24,25]. By modifying four amino acids, we reproduced the human HEAT repeat 15 in *C. elegans* SFTB-1. As a result, worms became sensitive to pladienolide B and herboxidiene but maintained resistance to sudemycin D6. Further humanization of SFTB-1 and other SF3B complex components would

likely increase the sensitivity to these drugs and expand the number of known splicing modulators effective in *C. elegans*. As an example, while PHF5A is highly conserved in *C. elegans*, the Y36 position, which confers resistance to PB, HB, and sudemycin D6 when mutated [25], corresponds to a histidine. Thus, the H36Y substitution in *C. elegans* might increase sensitivity to splicing modulators.

It is worth mentioning that this is the first report of an invertebrate animal model sensitive to PB and HB. Previously, splicing inhibition by PB *in vivo* was reported in yeast strains that were edited to humanize some Hsh155 HEAT repeats and mutated to block PB efflux [26]. As an added value to the *C. elegans* toolkit, we have generated a strain that presents WT characteristics but is sensitive to splicing modulators such as PB and HB, which we believe has great potential for splicing-related studies in this nematode.

*SF3B1* mutations are mutually exclusive with other splicing factor mutations present in tumors. *U2AF1* and *SRSF2* missense mutations are reported in 11% and 12–15% of MDS, respectively [19], and very rarely overlap with *SF3B1* mutations, which are present in 17–28% of MDS patients [37,39]. Hence, cells may not handle all the AS defects produced by the combination of mutations in any of these three proteins. To further validate our model as a tool for identifying vulnerabilities in cells with compromised SF3B1 function, we mimicked *U2AF1*[S34F] and *SRSF2*[P95H] mutations in *C. elegans* and detected a functional interaction between both mutants and the *sftb-1*[Q552P, R643C, K718E] allele. Additionally, *C. elegans* *rsp-4* and *uaf-2* missense mutations could be used for RNA-seq analyses and RNAi screens to study the consequences and vulnerabilities of these mutations, respectively.

We have demonstrated that *C. elegans* can be a valuable model to explore vulnerabilities of cells with mutations in *SF3B1*, although the molecular consequences of *sftb-1* mutations were different from the ones observed in other organisms. Replacement of several HEAT repeats in yeast Hsh155 with their human counterparts is functional and supports splicing [60]. Similarly, partial or total replacement of *sftb-1* with its human ortholog could result in a more accurate model. To date, only one human gene has been functionally transplanted to *C. elegans* [61], in part due to low efficiencies of inserting long DNA fragments by CRISPR. Such technical difficulties can now be bypassed by the use of Nested CRISPR [62]. Thus, humanized *sftb-1* nematodes obtained by this method would be very efficient as *in vivo* models to further understand the impact of *SF3B1* mutations in cancer and as a platform for large-scale genetic and pharmacological screens.

## Materials and methods

### *Caenorhabditis elegans* strains

*C. elegans* strains were maintained using standard methods [63,64] at the specified temperature. Before conducting the experiments, strains were grown for at least two generations at the experimental temperature. Unless otherwise specified, we used Bristol N2 as the WT strain. Mutant strains with single *sftb-1* missense mutations (CER218, CER220, CER236, and CER238) were outcrossed x2. WT siblings resulting from the second outcross were given specific strain names (CER217, CER221, CER237, and CER239, respectively) and kept for closer comparison. CER351, CER442, and CER450 strains were outcrossed x2. In order to maintain the *cer6* deletion, heterozygous mutants were crossed with the EG7893: *oxTi615 unc-119(ed3) III* strain so that a red fluorescent marker was inserted 0.15 cM away from the *sftb-1* WT locus. To study the lasting of maternally inherited SFTB-1, we made the compound heterozygous strain CER505: *sftb-1(cer6/cer114[mCherry::sftb-1]) III*. All the strains used in this study are listed in S5 Table.

## Protein alignments

*H. sapiens* SF3B1 (Uniprot: O75533) and *C. elegans* SFTB-1 (Uniprot: G5EEQ8) protein sequences were aligned using the T-Coffee algorithm [65] with default settings. S1 Fig was generated with Jalview [66].

## CRISPR/Cas9 mutant and reporter strains

Specific guide RNAs were designed using both *Benchling* (www.benchling.com) and *CCTop* [67] online tools. All CRISPR/Cas9 mutant and reporter strains were obtained following a co-CRISPR strategy [68] using *dpy-10* as a marker to enrich for genome-editing events. In all cases, mixes were injected into gonads of young adult $P_0$ hermaphrodites using the Xeno-Works Microinjection System and following standard *C. elegans* microinjection techniques. $F_1$ progeny was screened by PCR using specific primers and $F_2$ homozygotes were confirmed by Sanger sequencing. As we optimized the protocol for CRISPR/Cas9 genome editing in our laboratory while preparing this manuscript, different reagents and conditions were used to produce the distinct mutants. The *sftb-1(cer6)* deletion mutant allele was obtained by nonspecific double-strand break repair on our first attempt to generate the K718E mutation. Only in this experiment, guide RNAs and Cas9 were expressed from plasmids that were purified with the NucleoBond Xtra Midi Kit (Macherey-Nagel). In the remaining mixes, ribonucleoprotein complexes containing crRNA, tracrRNA, and Cas9 were annealed at 37˚C for 15 minutes prior to injection. All the reagents and injection mixes used in this study are listed in S6 Table.

## *C. elegans* microscopy

Worms were mounted on 2% agar pads with 10 mM tetramisole hydrochloride and imaged using a ZEISS Axio Observer Z1 inverted fluorescence microscope. Images were processed using ZEISS ZEN 2012 (blue edition) software and Fiji.

## Gonad dissection and DAPI staining

Worms expressing mCherry::SFTB-1 were anesthetized with 0.33 mM tetramisole hydrochloride and their heads cut using a pair of 25-gauge needles to extrude gonad arms. Released gonads were fixed in 4% paraformaldehyde in PBS for 20–30 minutes and washed with PBS + 0.1% Tween 20 for 10 minutes. After performing 3 washes, gonads were transferred to a microscope slide and mounted using 10 µl DAPI Fluoromount-G (SouthernBiotech). Images were acquired the same day to avoid mCherry signal loss.

## *sftb-1* deletion mutant larval arrest quantification

Synchronized L1-arrested $F_2$ worms from heterozygous *sftb-1(cer6)/oxTi615* $P_0$s were seeded onto NGM plates with OP50 and imaged after 48 hours of growth at 20˚C using a ZEISS Axio Observer Z1 inverted fluorescence microscope at 10X magnification. Body length was measured manually by drawing the midline of the worm using the freehand line tool in Fiji.

## Phenotypic characterization

Similar experiments were conducted to characterize *sftb-1*[Q552P, R643C, K718E], *rsp-4*[P100H], and *uaf-2*[S42F] mutant phenotypes. Briefly, synchronized L1-arrested worms of the specified genotype were first seeded onto NGM plates with OP50 and, after a few hours to allow recovery from the L1 arrest, 60 animals per genotype were separated individually to 12-well NGM plates seeded with OP50. Plates were incubated at 20˚C (Figs 4 and S3) or 25˚C (Fig 1F–1H). The developmental stage of individual worms was assessed visually every 24

hours under a stereomicroscope based on animal size and germline morphology. Phenotypes were scored at day 4. In order to determine brood sizes of *sftb-1*[Q552P, R643C, K718E] and WT animals, a proportion of worms were transferred into new 35-mm NGM plates at day 4 and subsequently moved to fresh plates every 12–24 h until egg laying ceased. Progeny were counted 36–48 h after passages.

### RNA isolation

To study the transcriptional consequences of the K718E mutation, we used two homozygous mutant lines (CER218 and CER220) derived from two independent CRISPR/Cas9 events (*cer3* and *cer7* alleles, respectively), and the corresponding WT siblings (CER217 and CER221). Synchronized L4 animals were collected in M9 after growing for 41 hours at 20˚C and total RNA was isolated using TRI Reagent (MRC, Inc.) and the PureLink RNA Mini Kit (Ambion), following the manufacturer's instructions.

*sftb-1(cer6)* homozygous larvae were obtained by automatically sorting non-red animals from a synchronized CER191 population grown at 20˚C for 23 hours, using the COPAS Biosorter system (Union Biometrica). We conducted two biological replicates with an average of 1.6% non-red recombinants and 5.25% red contaminants in the samples. The same size and fluorescence window were used to sort two N2 WT populations collected under the same experimental conditions. Two CER276 biological replicates were collected independently without sorting, after growing for 23 hours at 20˚C. Total RNA from the six samples was extracted using TRI Reagent, followed by aqueous and organic phase separation using 1-Bromo-3-chloropropane (Sigma-Aldrich), and RNA was precipitated with 2-propanol. After treatment with TURBO DNase (Thermo Fisher Scientific), RNA was purified following a phenol/chloroform extraction protocol.

In all cases, total RNA was quantified by Qubit RNA BR Assay kit (Thermo Fisher Scientific) and the integrity was checked by using the Agilent RNA 6000 Nano Kit (Agilent).

### RNA sequencing

Four RNA-seq libraries (from CER217, CER218, CER220, and CER221 strains) were prepared with the KAPA Stranded mRNA-Seq Illumina Platforms Kit (Roche-Kapa Biosystems) following the manufacturer's recommendations. Briefly, 500 ng of total RNA was used as the input material, the poly-A fraction was enriched with oligo-dT magnetic beads and the mRNA was fragmented. Strand specificity was achieved during the second strand synthesis performed in the presence of dUTP instead of dTTP. The blunt-ended double stranded cDNA was 3´adenylated and Illumina indexed adapters from TruSeq™ Stranded Total RNA Sample Preparation Kit with Ribo-Zero Gold (Illumina) were ligated.

The remaining six samples (two biological replicates from N2, CER276, and homozygous *sftb-1(cer6)* larvae) were prepared using the TruSeq™ Stranded Total RNA kit protocol (Illumina). Briefly, rRNA was depleted from 500 ng of total RNA using the Ribo-Zero Gold rRNA Removal Kit and fragmented by divalent cations, with a major peak at 160 nt. Following the fragmentation, first and second strand synthesis was performed, also ensuring the library directionality. The cDNA was adenylated and ligated to xGen Dual Index UMI Adapters (IDT) for paired-end sequencing.

Both types of library ligation products were enriched with 15 PCR cycles and the final library was validated on an Agilent 2100 Bioanalyzer with the Agilent DNA 1000 Kit (Agilent).

The libraries were sequenced on HiSeq 2000 (Illumina) in paired-end mode with a read length of 2x76 bp using the HiSeq SBS Kit V4 50 cycle kit in a fraction of a sequencing v4 flow cell lane (HiSeq PE Cluster Kit V4—cBot). Image analysis, base calling, and quality scoring of

the run were processed using the manufacturer's software Real Time Analysis (RTA 1.18.66.3) and followed by generation of FASTQ sequence files.

## Bioinformatic analysis

RNA-seq reads were mapped against the *C. elegans* reference assembly (WBcel235.87) with STAR [69]. Genes and transcripts were quantified with RSEM [70]. Differential expression analysis was performed with DESeq2 R package [71], and detection of differential alternative splicing events was done with rMATS [33].

## RNAi screen

Information about the selected splicing genes was collected from SpliceosomeDB [72] or from the literature. RNAi clones were obtained from the ORFeome library [73] or the Ahringer library [28] and clone insert size was validated by PCR. For screening, we used 24-well plates containing NGM supplemented with 50 µg/ml ampicillin, 12.5 µg/ml tetracycline and 3 mM IPTG. Each well was seeded with 80 µl of bacterial culture and dsRNA expression was induced overnight at room temperature. WT and mutant worms were tested in duplicates. Between 10 and 20 worms synchronized at the L1 stage were seeded in each well, and phenotypes were scored visually at the adult stage (72–96 h post-seeding at 25˚C). Bacteria expressing dsRNA against *gfp* were used as negative control. Screens in *sftb-1*[K718E] and *sftb-1*[R643C] backgrounds were performed twice, at two different times, with different batches of plates and RNAi cultures, and scored by two different persons. Several candidates were selected for further validation, but the existence of false negatives in the screen is possible due to the noisy nature of RNAi for essential genes in small plates.

## Interaction between *teg-4* and *sftb-1* mutations

Worms of the indicated genotype were singled onto individual 35-mm NGM plates with OP50 at the L4 stage and transferred to new plates every 12 hours until all eggs were laid. Numbers of dead embryos and hatched larvae were scored 24 and 48 hours after passages, respectively. Since the double mutant *sftb-1*[K718E]; *teg-4(oz210)* could not be maintained in homozygosis, $F_1$ worms from *sftb-1*[K718E]; *teg-4(oz210/+)* $P_0$ worms, including homozygous and heterozygous animals for *teg-4(oz210)*, were picked at the L4 stage and their progeny scored. These $F_1$s were genotyped after conducting the experiment to include only *sftb-1*[K718E] and *sftb-1*[K718E]; *teg-4(oz210)* data.

## Validation of candidates from the RNAi screen

In all cases, we used 55-mm or 35-mm NGM plates supplemented with 50 µg/mL ampicillin, 12.5 µg/mL tetracycline and 3 mM IPTG, seeded with specific RNAi bacterial clones. RNAi targeting *gfp* was used as negative control and experiments were performed at 25˚C.

In Fig 3C and 3D and S6B Fig, synchronized L1-arrested worms of the indicated genotypes were seeded onto 55-mm plates, and imaged directly on the plates after 48 hours of growth using a stereomicroscope. Body length was determined automatically with WormSizer [74]. In Fig 3E–3G, synchronized L1-arrested animals were seeded onto 55-mm plates and separated into individual 35-mm plates at the L4 stage. Worms were transferred every 24 hours and the presence of dead embryos was determined 24 hours after passages. In S6A Fig, synchronized L1-arrested animals were seeded onto 55-mm plates and separated into individual wells in 24-well RNAi plates at the L4 stage, and the fertility phenotypes assessed in 2-day old adults.

## Number of germ cells assay

WT or *sftb-1(cer6)* homozygous worms were collected at the indicated time points, fixed using Carnoy's solution (60% absolute ethanol, 30% chloroform, 10% acetic acid), and washed three times in PBS + 0.1% Tween 20. Whole worms were transferred to a microscope slide and stained with DAPI Fluoromount-G in order to visualize and count germ cell nuclei.

## Maternal product in *sftb-1(cer6)* homozygotes

In order to monitor the presence of mCherry::SFTB-1 maternal product, *sftb-1(cer114)* or *sftb-1(cer6/cer114)* animals were separated blindly and allowed to lay eggs for 2, 9, and 12 hours in three separate plates in order to semi-synchronize the $F_1$ populations. Thus, only a fraction (23 hours) of the whole egg-laying period was covered. Red expression levels were assessed at different moments in order to include all the stages represented, which were designated visually based on anatomic features. *sftb-1(cer6/cer114)* animals were distinguished from *sftb-1(cer114)* animals by the presence of non-red arrested larvae in the next generation.

## Semi-quantitative reverse-transcriptase PCR

N2, CER442, and CER450 animals grown at 20°C were collected at the L2 (24 hours after seeding L1-arrested worms) and adult stages. Total RNA was extracted using TRI Reagent, followed by aqueous and organic phase separation using 1-Bromo-3-chloropropane (Sigma-Aldrich), and RNA was precipitated with 2-propanol. After treatment with DNase I (Thermo Fisher Scientific), cDNA was synthetized with the H Minus First Strand cDNA Synthesis Kit (Thermo Fisher Scientific) following the manufacturer's instructions. Primer sequences are available upon request.

## Spliceosome inhibitors experiments

Experiments in liquid media were conducted in 96-well plates. Experiments in S4 Table were conducted as follows: 25 μl of dead OP50 resuspended in a mix of freshly prepared S Medium [64], 4 μg/ml cholesterol, 250 μg/ml streptomycin, and 62.5 μg/ml tetracycline were added to each well. Subsequently, 25 μl 2.4X drug and 10 μl L1-arrested worms at a density of 1–2 worms/μl were added to each well. Each condition was tested in duplicate in the same 96-well plate. Plates were incubated in a humid chamber with gentle agitation at room temperature for the indicated time period. 25 μl water (experiments 1–3) or 0.5% DMSO (Sigma-Aldrich) (experiments 4 and 6) were used as negative controls. In experiment 5, 50 μM sudemycin D6 or 0.5% DMSO were injected into the gonads or intestine of young adult hermaphrodites using standard microinjection techniques, and injected animals were recovered in individual 35-mm NGM plates with OP50 and incubated at 20°C.

The protocol for liquid experiments in Fig 5C and 5D was slightly modified. In brief, 64.5 μl S Medium supplemented with 5 μg/ml cholesterol, 50 μg/ml streptomycin, and 50 μg/ml ampicillin were added to each well. This supplemented S Medium was used to resuspend a pellet of dead OP50 to an optical density of 1.1–1.2, and 25 μl of bacteria were dispensed to each well. Subsequently, 10 μl L1-arrested worms at a density of 4–6 worms/μl, and 0.5 μl 200X drug were added to each well. Each condition was tested in triplicate in the same 96-well plate. Plates were incubated in a humid chamber at 25°C.

## Statistical analyses

In all figure legends, 'N' denotes the number of independent replicate experiments performed, while 'n' indicates the total number of animals analyzed in each condition (different genotypes

are separated by commas). Statistical analyses were performed in GraphPad Prism 6 and R. Statistical tests used are reported in the figure legends. Figs 1A, 1D, 1F–1H, 2B–2D, 3B–3G, 4, 5C and 5D, and some panels in S2, S3, S4, S5 and S6 Figs were generated by using the ggplot2 R package.

## Supporting information

**S1 Fig. Most of the SF3B1 mutated residues are conserved in the *C. elegans* ortholog SFTB-1.** Protein sequence alignment of human SF3B1 (Hs) and worm SFTB-1 (Ce). The yellow histogram and the numerical index represent the degree of conservation (* corresponds to the maximum degree of conservation). U2AF ligand motifs (ULMs), p14-interacting region and HEAT domains are indicated in boxes. A cross symbol (X) below the conservation score indicates positions where missense mutations have been reported in human cancers (COSMIC database) at different frequencies. Residues that are relevant for this study are highlighted in green.
(TIF)

**S2 Fig. *cer6* homozygotes inherit SFTB-1 maternal product.** (A) Schematic representation of the *sftb-1* locus. Gray boxes indicate exons, connecting lines indicate introns. Left and right orange boxes represent predicted 5' and 3' untranslated regions, respectively. In pink, 29 nucleotides that are deleted in the *cer6* allele. Predicted amino acids resulting from translating the WT or *cer6* alleles are shown below the DNA sequence. The frameshift caused by the deletion introduces a premature stop codon in the *cer6* transcript (colored in red). Scale bar, 100 base pairs (bp). (B) Germ cell number of WT (N2 strain) or homozygous *sftb-1(cer6)* synchronized larvae grown for 14, 23, and 40 hours at 20˚C. In WT worms, differentiating spermatocytes were not included at the last time point, and the number of germ cells was counted in one gonad arm only. Black and gray lines connect the median germ cell number at each time point in WT and *sftb-1(cer6)* worms, respectively. The dotted line represents the four germ cells that are normally present in recently hatched L1 larvae; these cells will resume proliferation during the L1 stage. Between 2 and 10 animals per condition were scored (N = 1). (C) Schematic representation of the compound heterozygous strain CER505: *sftb-1(cer6)/sftb-1 (cer114[mCherry::sftb-1]) III*. (D) Total number of progeny laid by heterozygous (*cer6/+*) or WT (*+/+*) worms from a CER505 population (n = 7, 9; N = 1). The number of $F_1$ larvae expressing mCherry::SFTB-1 was not significantly different between both groups (n.s.), while *cer6/+* animals laid a significantly higher number of dead embryos which lacked red fluorescence (Mann-Whitney's test; * p<0.05). Heterozygotes also segregated a number of non-red larvae that were arrested and were not observed in WT plates, indicating that they were *cer6/cer6* animals. '+' denotes the *cer114[mCherry::sftb-1]* allele. Dots represent measures in individual worms, overlaid to Tukey-style boxplots. (E) Mean percentage of non-red $F_1$ arrested larvae (% Lva), F1 dead embryos (% Emb), and mean brood size in worms of the indicated genotypes, from data represented in panel D. (F) mCherry::SFTB-1 expression in $F_1$ progeny from wild-type (top, n = 3) or heterozygous (bottom, n = 9) animals from a CER505 population (N = 1). At each stage, vertical lines correspond to individual worms, and solid red, open red, and black dots represent the total number of red, dim red, or non-red progeny observed, respectively. Representative DIC and fluorescence microscopy images are shown below the graph (scale bar, 25 μm). The mean percentage of dim red and non-red progeny laid by heterozygotes is indicated in late stages and is not specified in early stages as it was 0%.
(TIF)

**S3 Fig. Molecular design of *sftb-1* cancer-related mutations and *sftb-1*[Q552P, R643C, K718E] phenotypes at 20°C.** (A) Schematic representation of the *sftb-1* locus, as in S2 Fig. The position of the triplets encoding the three mutated residues in this study (Q552, R643 and K718) is indicated. Scale bar, 100 bp. (B) Molecular details of the three *sftb-1* missense mutations generated by CRISPR/Cas9. In WT alleles, the crRNA sequence used is indicated by orange nucleotides, the protospacer-adjacent motif (PAM) is underlined and the Cas9 cut site is indicated with a black arrowhead. In mutant alleles, synonymous mutations introduced to prevent partial recombination events and to improve primer specificity for mutant allele detection by PCR are indicated in blue. Mutated codons and the corresponding amino acids are colored in pink. (C) Bars represent mean incidence of different phenotypes observed in *sftb-1* [Q552P, R643C, K718E] worms at 20°C, while dots represent percentages observed in each replicate (n = 116, 109; N = 2). (D) Brood size of WT or *sftb-1*[Q552P, R643C, K718E] worms at 20°C (n = 33, 46; N = 2). Dots represent values for each individual animal, overlaid to Tukey-style boxplots. (E) *sftb-1*[Q552P, R643C, K718E] animals present a mild developmental delay at 20°C. Dot sizes represent the proportion of the population at each stage (n = 116, 109; N = 2). Statistics: (C), Fisher's exact test (D), Student's t-test (unpaired, two-tailed). n.s., no significant difference, **** $p < 0.0001$.
(TIF)

**S4 Fig. Pathway enrichment analysis of genes with differentially expressed transcripts in different *sftb-1* mutant strains.** Significantly enriched functional terms in the distinct datasets are listed (padj<0.01). Bars represent the adjusted enrichment p-values in negative $\log_{10}$ scale, color-coded by mutant strain. Solid and clear bars denote enriched terms in upregulated and downregulated genes, respectively. The number of genes with differentially expressed transcripts belonging to each category is shown. The analysis was performed with the g:GOSt tool in g:Profiler, and only biological pathways from KEGG and WikiPathways databases are shown.
(TIF)

**S5 Fig. Common AS events deregulated by *sftb-1* alleles.** (A) Summary of the overlapping of genes with AS changes (left) and the overlapping of AS events (right) between groups. (B) Venn diagram displaying the overlapping AS events between the three datasets. The total number of significant events with inclusion level difference > 0.1 in each dataset is indicated. (C) Plot showing the inclusion level difference (InclLevelDifference) of AS events that were deregulated in at least two datasets ('common AS events'), color-coded according to the dataset. Different genes are represented along the x-axis, and events are separated by event type. Some genes (*acdh-2*, *fbn-1*, *Y69H2.3*, *unc-89*, *swp-1*, and *ate-1*) shared more than one event in different datasets.
(TIF)

**S6 Fig. *sftb-1(cer6)* heterozygotes are sensitive to RNAi of five interactors.** (A) *mog-2 (RNAi)*, *smr-1(RNAi)*, and *teg-4(RNAi)* cause a significant mild increase in sterility in *sftb-1 (cer6)* heterozygotes compared to WT at 25°C (n≥56; N = 2; * $p < 0.05$, ** $p < 0.01$, **** $p < 0.0001$; Fisher's exact test). Red dots indicate percent sterility observed in each replicate. Gray bars, $P_0$-treated worms giving rise to <5 $F_1$ larvae and some dead embryos ('dead progeny' category); black bars, $P_0$-treated worms that laid neither larvae nor dead embryos ('no progeny' category). (B) *sftb-1(RNAi)* and *uaf-2(RNAi)* induce an earlier larval arrest in *cer6*/ + heterozygous worms. Dots represent individual worm lengths after 48 h of RNAi treatment at 25°C, overlaid to Tukey-style boxplots (n≥424; N = 2; **** $p < 0.0001$; Kruskal-Wallis test with Dunn's multiple comparison test). The genotype of the strain used for these experiments

was CER190: *sftb-1(cer6)/dpy-17(e164) unc-79(e1068) III*.
(TIF)

**S7 Fig. Effect of *uaf-2*[S42F] and *rsp-4*[P100H] mutations on a subset of AS events.** (A) cDNA samples from WT (N2), *uaf-2*[S42F], and *rsp-4*[P100H] worms at L2 and adult stage were used to analyze the presence of alternative isoforms by semiquantitative (sq) RT-PCR (N = 1). The alternative exons are numbered and shaded in gray in the schematic representation on the right, together with upstream and downstream exons where the forward and reverse primers annealed, respectively. The three distinct isoforms in *unc-32* E4 are indicated with red lines on the left. *act-1* was used as an endogenous control. (B) The 3'SS sequence of the alternative exons is indicated, as well as the CCNG and GGNG motifs found in alternative exons using FIMO with a p-value < 0.01. (C) Technical replicate of *tos-1* E1-E2 and *tos-1* E1-E4 sqRT-PCRs to verify that *uaf-2*[S42F] promotes intron 1 retention (slowest migrating band in both gels) and exon 3 skipping (fastest migrating band in bottom gel), both indicated with blue boxes. 'E' indicates exon, while 'I' indicates intron.
(TIF)

**S1 Table. Alternative splicing analysis of *sftb-1* mutants by rMATS.**
(XLSX)

**S2 Table. Differential expression analysis of *sftb-1* mutants by DESeq2.**
(XLSX)

**S3 Table. Information and phenotypes of the RNAi splicing screen.**
(XLSX)

**S4 Table. First assays with spliceosome inhibitors.**
(XLSX)

**S5 Table. *C. elegans* strains used in this study.**
(XLSX)

**S6 Table. CRISPR reagents and injection mixes.**
(XLSX)

## Acknowledgments

We acknowledge the members of the Cerón Laboratory for helpful discussions and comments on the manuscript. We also thank Alan Zahler and Sol Katzman for their advice in transcriptomic analyses. The *teg-4(oz210)* strain was kindly provided by Dave Hansen. Pladienolide B and sudemycin D6 were gifts from Juan Valcárcel. We thank CERCA Program / Generalitat de Catalunya for their institutional support.

## Author Contributions

**Conceptualization:** Julián Cerón.

**Formal analysis:** Xènia Serrat, Anna Esteve-Codina.

**Funding acquisition:** Julián Cerón.

**Investigation:** Xènia Serrat, Dmytro Kukhtar, Eric Cornes, Julián Cerón.

**Methodology:** Xènia Serrat, Dmytro Kukhtar, Eric Cornes, Helena Benlloch, Germano Cecere.

**Supervision:** Julián Cerón.

**Writing – original draft:** Xènia Serrat, Julián Cerón.

**Writing – review & editing:** Xènia Serrat, Julián Cerón.

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
