## [Decision Letter · Decision Letter 0]

2 Aug 2019

Dear Dr Ceron,

Thank you very much for submitting your Research Article entitled 'CRISPR editing of sftb-1/SF3B1 in C. elegans allows the identification of synthetic interactions with cancer-related mutations and the chemical inhibition of splicing' to PLOS Genetics. Your manuscript was fully evaluated at the editorial level and by independent peer reviewers. The reviewers appreciated the attention to an important problem, but raised some substantial concerns about the current manuscript. Based on the reviews, we will not be able to accept this version of the manuscript, but we would be willing to review again a much-revised version. We cannot, of course, promise publication at that time.

If you decide to revise the manuscript for further consideration at PLOS Genetics, please aim to resubmit within the next 60 days, unless it will take extra time to address the concerns of the reviewers, in which case we would appreciate an expected resubmission date by email to plosgenetics@plos.org.

[LINK]

We are sorry that we cannot be more positive about your manuscript at this stage. Please do not hesitate to contact us if you have any concerns or questions.

Yours sincerely,

H. Leighton Grimes

Associate Editor

PLOS Genetics

Gregory P. Copenhaver

Editor-in-Chief

PLOS Genetics

Reviewer's Responses to Questions

**Comments to the Authors:**

Reviewer #1: This is an interesting and well-performed study generating several models of cancer-associated spliceosomal gene mutations in C. elegans and identifying genetic and pharmacologic synthetic lethal interactions in these models. The studies are novel and experiments are well performed. Nonetheless, the addition of a few experiments aimed at getting deeper into mechanistic basis for some of the associations shown would help the manuscript. In addition, there are some improper references and citations that should be corrected as noted below:

-What is the basis for the synthetic lethal effects of mutations in SF3B1 and U2AF1/SRSF2 found here? While this synthetic lethal interaction has been shown in mice, identifying a molecular basis for this interaction was challenging and not very clear.

-Why are the humanized SF3B1 mutant worms sensitive to Pladienolide B but insensitive to Sudemycin D6? These compounds physically interact with several components of the SF3b complex (not just SF3B1). For example, several amino acid residues in PHF5A are required for drug binding to Pladienolide B. It would be important to note whether these proteins are conserved in sequence between C. elegans and humans.

-What are the effects of mutations in U2AF1S34P and SRSF2P95H on splicing? These mutations have been reported to impact splicing based on exonic sequences and/or sequences at the 3’ splice site (genomic regions which may be more conserved across species).

-In the Abstract, the following points should be corrected:

---SF3B1 mutations have not been shown to impart a growth advantage on their own. In fact, in most studies introduction of mutant SF3B1 actually imparts a growth disadvantage to human and mouse cells. This point alone has been a conundrum related to the study of these mutations.

---Cancer-associated SF3B1 mutations have not been shown to be deleterious when made homozygous in any animal models to date. This has been shown for SRSF2 however (Lee et al. Nature Medice 2016; PMCID PMC4899191). This paper would be important to cite here. SF3B1 mutations have been shown to be lethal when made hemizygous (with ablation of the wild-type allele in the presence of the mutant allele; Zhou et al. A chemical genetics approach for the functional assessment of novel cancer genes. Cancer Research 2015). This paper should also be cited.

-In the Author Summary, the following points should be corrected:

---The phrase “visual phenotype” should be edited to be more precise. I believe the authors mean “overt” phenotype (as opposed to a phenotype related to vision) but this needs to be clearly stated.

---The term “slight” when referencing alterations in splicing should be deleted. First, the splicing of one essential gene in a deleterious way can have massive effect on cells/organisms so the phrase “slight” is not appropriate.

---Pladienolide B binds to both wild-type SF3B1 protein as well as cancer-associated SF3B1 mutants. Thus, the phrase “hit the wild-type copy” is not appropriate (“copy” also suggests the genomic allele and not the protein).

-The comment in the Discussion about effects of SF3B1 K700E mutations on splicing in mouse versus human cells should note the findings from Lee et al. 2018 reference cited here where a much greater overlap in aberrant splicing events between mouse and human SF3B1 K700E mutant cells was identified than in Obeng et al. or Mupo et. al papers listed (Obeng & Mupo et al. studies missed some very clearly overlapping splicing events across mouse & human SF3B1 K700E mutant cells).

Reviewer #2: In this manuscript, Serrat and colleagues design a series of transgenic C. elegans animals carrying cancer-associated mutations in several orthologs of splicing factor genes. The majority of the paper focuses on mutations in sftb-1, the worm ortholog of SF3B1, but they also study mutations introduced in uaf-2/U2AF1 and rsp-4/SRSF2, two additional splicing regulators. The authors have used these transgenic animals to look at phenotypes associated with these cancer-associated mutations. They demonstrate that although individual mutations in sftb-1 do not have dramatic phenotypic effects (unlike in humans), combining several mutations at the same locus do have pronounced effects on animal health and physiology. The authors then explore transcriptome changes in several of these mutant strains by RNA-seq analysis, and find that splicing patterns and gene expression patterns differ between mutants, with severity of phenotype correlating with the extent of transcriptome alterations. Next, the authors perform a focused RNAi screen to look at genetic interactions between alleles of sftb-1 and other splicing related factors, and identify 27 candidate modifier genes and validate one of these interactions with a partial loss of function allele. The authors extend their analysis of genetic interactions by making double mutant animals with other splicing factor genes containing cancer-associated mutations (sftb-1; uaf-2 and sftb-1; rsp-4), and find synthetic interactions between these alleles as well. Finally, the authors also test the responsiveness of a partially humanized version of sftb-1 to pladienolide B, and demonstrate that these transgenic animals display a dose-sensitive phenotype.

Overall, I found that this manuscript has several very interesting results and conceptual advances, including the establishment of a worm strain where splicing can be modulated by a chemical. Although the authors do not delve deeper into any of the interesting genetic interactions uncovered or further pursue experiments related to their humanized sftb-1 mutant that responds to PB, the work presented will pave the way towards additional interesting lines of inquiry. As such I think the work is suitable for publication, but it would be nice to have more clarification/additional data included to strengthen a few points:

1) In the RNAi screen, it appears that among the 27 hits, the authors score an interaction as more severe relative to a sibling matched wild type from their outcrossing experiments (Table S3). While I appreciate that the authors are trying to use matched sibling wild type strains to account for background effects, the differences in the phenotypes scored for the wild type strains (CER221, CER237, and CER239) can be somewhat variable between experiments. This variability leads to some concerns that there may be variability in the depletion of the genes targeted by RNAi from experiment to experiment rather than variability due to background genetic modifiers (I would be very surprised to see this much variability due to off-target effects due to CRISPR editing for example).

Minimally, it would be nice to see more information on the details of the RNAi screens:

-How many technical/biological replicates were scored

-Were phenotypes called by different individuals or the same individual

-Were all experiments performed together (i.e same batch of plates / RNAi bacteria) or were different control strains tested on different days?

This information would give the reader a bit more clarity in terms of how much confidence to place in some of the preliminary RNAi screen data. It is nice that the teg-4 example validates well by additional genetic analysis.

2) For the transcriptome analysis, it would be nice to know the overlap between the transcriptome changes detected in the different mutants? Either there is a lot of overlap suggesting that the mutations lead to similar effects on sftb-1 activity, or there is not so much overlap suggesting that different mutations have different consequences. Either result is interesting. Perhaps a Venn Diagram or a heatmap would give the reader a better idea of the extent of overlap between targets and direction of change.

3) Are the target transcripts affected by these sftb-1 mutations enriched in any functional categories? Have the authors performed a GO analysis? It would be interesting to hear the results of such an analysis.

Reviewer #3: The manuscript by Serrat et al. entitled “CRISPR editing of sftb1/SF3B1 in C. elegans allows the identification of synthetic interactions with cancer-related mutations and the chemical inhibition of splicing.” describes the generation and characterization of novel C.elegans mutants in the splicing factor sftb1/SF3B1. The authors describe synthetic lethal interplay between disease-associated sftb1 alleles and mutations in other splicing factors. Moreover, they show that humanizing C.elegans sftb1 sensitizes mutant worms to treatment with the splicing modulator Pladienolide B. Consistent with work in human and murine cells, the authors demonstrate that the disease-associated sftb1 alleles are more sensitive to Pladienolide B treatment. Together, the paper demonstrates the utility of these new animal models to identify novel synthetic lethal interaction with sftb1 mutations. Overall the experiments are sound and well controlled, but a few points need additional clarification.

1. The authors state that the sftb1 loss-of-function mutants are viable until larval stages due to residual maternal contributed wild-type product. Please include quantification of this residual maternal material through embryonic and early larval stages. It is important to understand the requirement for wild-type sftb1 for survival.

2. Knockdown of sftb1 was shown to be synthetic lethal with sftb1 (Q55P, R643C, K718E) mutants. Studies in murine models of other splicing factors suggest that cells heterozygous with null alleles are less affected than those heterozygous with disease-associated alleles. It would be of value to the community to know if this is also true for SF3B1. The genetic interactions should also be performed with sftb1 null/wild type worms.

3. There are similar types of splicing changes in the various sftb1 mutants, but are the same transcripts/genes affected. Quantify the overlap of the exact splicing changes across the RNA-seq datasets and add 1-2 sentences commenting on the overlaps.

4. Out of the 104 RNAi clones tested, 27 showed a synthetic interaction with the sftb1 mutants. Is there anything special about the 27 compared to the 104? Do they act as a specific step of the splicing cycle? Are they enriched for SF3B1-interacting proteins? Please add 1-2 sentences commenting on the hits.

5. RNAi for teg4 is the strongest interactor with the sftb1 mutants, but it also shows major lethality in wild types. Add a sentence clarifying the effect is not overly selective to the sftb1 mutants.

6. By mutating four amino acids in sftb1, the authors were able to sensitize worms to pladienolide B but not sudemycin D6. Please add a sentence to the results or discussion commenting on the different binding sites for PB and sudemycin within SF3B1 that imparted this specificity.

**Have all data underlying the figures and results presented in the manuscript been provided?**

Reviewer #1: Yes

Reviewer #2: No: Large-scale transcriptome datasets do not have associated accesssion numbers. Presumably the authors will upload these to the NCBI upon acceptance

Reviewer #3: Yes

PLOS authors have the option to publish the peer review history of their article (what does this mean?). If published, this will include your full peer review and any attached files.

Reviewer #1: No

Reviewer #2: No

Reviewer #3: No

---

## [Decision Letter · Decision Letter 1]

5 Oct 2019

Dear Dr Ceron,

We are pleased to inform you that your manuscript entitled "CRISPR editing of sftb-1/SF3B1 in C. elegans allows the identification of synthetic interactions with cancer-related mutations and the chemical inhibition of splicing" has been editorially accepted for publication in PLOS Genetics. Congratulations!

Yours sincerely,

H. Leighton Grimes

Associate Editor

PLOS Genetics

Gregory P. Copenhaver

Editor-in-Chief

PLOS Genetics

Comments from the reviewers (if applicable):

Reviewer's Responses to Questions

**Comments to the Authors:**

Reviewer #1: The authors have thoughtfully addressed my initial questions and concerns. I have no further issues with the manuscript.

Reviewer #2: The authors have addressed my concerns and added some additional interesting data. I find the manuscript suitable for publication.

Reviewer #3: All concerns were addressed.

**Have all data underlying the figures and results presented in the manuscript been provided?**

Reviewer #1: Yes

Reviewer #2: Yes

Reviewer #3: Yes

PLOS authors have the option to publish the peer review history of their article (what does this mean?). If published, this will include your full peer review and any attached files.

Reviewer #1: No

Reviewer #2: No

Reviewer #3: No

**Data Deposition**

http://datadryad.org/submit?journalID=pgenetics&manu=PGENETICS-D-19-01147R1

**Press Queries**

---

## [Editor Report · Acceptance letter]

16 Oct 2019

PGENETICS-D-19-01147R1 

CRISPR editing of *sftb-1/SF3B1* in *Caenorhabditis elegans* allows the identification of synthetic interactions with cancer-related mutations and the chemical inhibition of splicing 

Dear Dr Ceron, 

We are pleased to inform you that your manuscript entitled "CRISPR editing of *sftb-1/SF3B1* in *Caenorhabditis elegans* allows the identification of synthetic interactions with cancer-related mutations and the chemical inhibition of splicing" has been formally accepted for publication in PLOS Genetics! Your manuscript is now with our production department and you will be notified of the publication date in due course.

With kind regards,

Matt Lyles

PLOS Genetics

On behalf of:
